# Progress in African Swine Fever Vector Vaccine Development

**DOI:** 10.3390/ijms26030921

**Published:** 2025-01-22

**Authors:** Yue Yang, Hengxing Yuan, Yulu Zhang, Ji Luan, Hailong Wang

**Affiliations:** State Key Laboratory of Microbial Technology, Institute of Microbial Technology, Helmholtz International Lab for Anti-Infectives, Shandong University–Helmholtz Institute of Biotechnology, Shandong University, Qingdao 266237, China; yangyue-@mail.sdu.edu.cn (Y.Y.); yuanhengxing@sdu.edu.cn (H.Y.); zhangyulu@mail.sdu.edu.cn (Y.Z.); luanji@sdu.edu.cn (J.L.)

**Keywords:** African swine fever, vaccines, ASFV antigens, vectors, cytokines

## Abstract

African swine fever (ASF) is a highly lethal, infectious, hemorrhagic fever disease, characterized by an acute mortality rate approaching 100%. It is highly contagious, and results in significant losses to the global hog industry as it spreads. Despite incremental progress in research on the African swine fever virus (ASFV), a safe and effective commercial vaccine has yet to be developed. Vector vaccines, a promising type of vaccine, offer unique advantages, and are a primary focus in ASFV vaccine research. This paper focuses on the characteristics of viral, bacterial, and yeast vector vaccines; elucidates the immunological mechanisms associated with antigens; lists the types of antigens that have significant potential; discusses the feasibility of using exogenously expressed cytokines to enhance the protective power of vector vaccines; and, finally, discusses the types of vectors that are commonly used and the latest advances in this field.

## 1. Introduction

African swine fever (ASF) is a highly lethal, infectious, hemorrhagic fever disease affecting pigs. It has been listed by the World Organization for Animal Health (WOAH) as a compulsory notifiable disease due to its rapid onset, brief disease course, and significant detrimental impact. The African swine fever virus (ASFV) poses a threat to pigs of all ages, with infected individuals often exhibiting symptoms such as high fever, a loss of appetite, cyanosis, and severe bleeding, leading to an acute mortality rate of nearly 100% [1]. ASFV is transmitted through direct contact with blood, urine, nasal secretions, saliva, and feces, as well as through close aerosol exposure [2]. In addition to this, feed contaminated with the virus is also a route of transmission [3]. Since its first discovery in Kenya, ASFV has spread to countries in Africa, Europe, the Americas, and Asia over the decades, demonstrating that ASFV is a global problem [4]. Studies on the control of ASF using vaccines are hampered by limitations in the exploration of antigenic ASFV genes, as well as a lack of scientific evidence on the pathogenesis of ASFV and a lack of an in-depth mechanistic understanding of the innate immune evasion strategy of ASFV. To date, no safe commercial vaccine has been developed, due to our limited knowledge of ASFV. The two vaccine candidates approved in Vietnam in 2023, AVAC ASF LIVE (strain ASFV-G-∆MGF) and NAVET-ASFVAC (strain ASFV-G-∆I177L), have both been discontinued due to biosafety concerns [5]. The developed viral vector vaccines have failed to provide adequate protection in pigs, suggesting that research in this direction requires a deeper understanding of ASFV. Controlling ASF remains a challenge, and there is an urgent need to develop a vaccine that provides complete protection [6].

For more than 40 years, scientists have employed various strategies to develop effective vaccines for ASF, including inactivated vaccines, live attenuated vaccines, subunit vaccines, nucleic acid vaccines, gene deletion vaccines, and vector vaccines [7]. While the traditional inactivated vaccine is effective in inducing antibodies and can block the virus in body fluids, it has limited efficacy in inducing specific cytotoxic CD8+T cells (CTLs) [8]. Consequently, the inactivated vaccine, which is effective against most pathogens, cannot provide adequate protection for pigs against ASFV infection. Understanding of protective immunity to ASFV remains unclear, which significantly hampers subsequent research on an ASF vaccine [9]. Conventional attenuated vaccines can only protect pigs against homologous viruses, and not against heterologous viruses [10]. Subunit and DNA vaccines can generate ASFV-specific antibodies and stimulate cellular immune responses; however, they do not provide complete protection [11,12]. In contrast, viral, bacterial, and yeast vector vaccines can both stably and efficiently elicit porcine humoral immune responses mediated by specific antibodies and cellular immune responses that secrete interferon-γ (IFN-γ). Additionally, these vaccines enhance safety levels, as virulence genes can be deleted through gene editing, or the virus may lose its ability to replicate. Consequently, vector vaccines represent a key focus for research and development in ASF prevention and control technologies.

## 2. Overview of ASFV

### 2.1. Genome

ASFV is a linear double-stranded DNA virus and the sole member of the *Asfivirus* genus within the *Asfarviridae* family. Notably, it is also the only arbovirus DNA virus [13]. ASFV has an icosahedral shape. The genomic size of ASFV varies among different virus samples, ranging from 170 to 193 kb, and contains between 150 and 167 open reading frames, encoding 54 structural proteins and over 100 non-structural proteins. The terminal region of the virus genome is covalently closed by a hairpin structure with incomplete base pairs, adjacent to which is a reverse array of various tandem repeats [14]. ASFV primarily infects porcine monocytes and macrophages, typically entering host cells via endocytosis or microendocytosis. Following internalization, DNA replication and viral particle assembly occur in the cytoplasm, after which the virus is released from the cell surface [15]. Once packaged, the virus remains infectious both intracellularly and extracellularly, exhibiting a high degree of survivability.

### 2.2. Genotype and Serogroup

Some genes in ASFV exhibit genetic diversity, suggesting that ASFV may employ various mechanisms to achieve this diversity and acquire new phenotypic characteristics. The genotypic identification of ASFV primarily relies on the *p72* gene [16]. ASFV’s major capsid protein (p72) gene, designated as *B646L*, was one of the first genetic targets for large-scale evaluation of the genetic diversity in ASFV. Based on partial gene sequencing of *B646L*, 24 ASF genotypes have been identified, and standard ASFV genotypic markers have been established. This genotyping approach can be utilized to trace the potential origin of the virus and differentiate it from closely related strains [17,18].

Identifying the different types of ASFV hinges on the following two key factors: (1) the stimulation of a type-specific (homologous) acquired immune response in living animals, which does not confer resistance to secondary infections caused by a different (heterologous) virus type; and (2) the induction of the erythrocyte adsorption phenotype in infected cells. Consequently, the hemadsorption inhibition assay is commonly employed to assess the serological cross-reactivity of different ASFV isolates in vitro [17] in order to distinguish the type of virus antigen, namely the serogroup. To date, eight ASFV serogroups have been identified and fully characterized, although it is possible that additional serogroups may exist [19].

## 3. Research Progress on ASFV Vector Vaccines

Vector vaccines are a class of vaccines that utilize gene-edited viral, bacterial, or yeast genomes as vectors. These vaccines incorporate the antigen genes of exogenous target viruses into non-essential regions of their vector genomes, thereby creating recombinant vectors that are capable of expressing both self- and exogenous antigen proteins simultaneously. Currently, the vectors that are primarily selected for the expression of ASFV antigen genes include adenoviruses, poxviruses, baculoviruses, herpes viruses, bacteria, and yeast. Similarly to DNA vaccines, vector vaccines deliver DNA into host cells, which then encode antigen proteins and trigger a series of immune responses mediated by antibodies, T helper cells (CD4+ T cells/Th cell), and cytotoxic T lymphocytes (CTLs, CD8+ T cells) [20].

The protective efficacy of ASFV vector vaccines after vaccination is affected by several factors, mainly the antigenic proteins selected to be introduced to the vector, but also the type of vector, and even the adjuvant. Other factors, such as antigen-induced immune memory, compatibility with vectors, and interactions with the host, also have an impact on the efficacy of the vaccine. In 2019, Lokhandwala et al. investigated the immune response and protection induced by two recombinant adenovirus vectors, Ad-ASFV I and Ad-ASFV II, using ASFV Georgia 2007 infection. They observed that Ad-ASFV II, when combined with two different adjuvants, BioMize0226 and ZTS-01, elicited distinct immune responses. Specifically, the presence of the adjuvant BioMize0266 caused ASFV II to induce more antigen-specific antibodies in the presence of the same antigens used in the experiment, whereas the adjuvant ZTS-01 was hypothesized to lead to disease enhancement by inducing antigen-dependent enhancement. This finding indicates that different adjuvants may provoke varied immune responses [21]. In addition, several studies have suggested that mucosal vaccination is more effective in triggering a protective mucosal immune response to prevent early viral infection and transmission than intramuscular injection. Following vaccination in the form of nasal spray, antigens are taken up by antigen-presenting cells in the nasopharynx and processed into antigenic peptides, which are subsequently translocated to lymphocytes, eliciting mucosal IgA and resident-memory B cell and T cell responses in the mucous membranes of the upper respiratory tract [22,23]. Compared with systemic-memory T cells, mucosal resident-memory T cells are more responsive and have a greater capacity for antigen binding, cytokine secretion, and cell killing [24]. The effects of adjuvants and delivery modes are not the focus of this article, but the main antigens and vector types will be highlighted below. Furthermore, considering that some studies have introduced the idea of inserting exogenous porcine cytokines to construct genetic recombinant vaccines, this article will also explore the feasibility of inserting exogenous cytokines into vectors to enhance their protective efficacy.

### 3.1. The Introduction of Major Antigens

#### 3.1.1. Immunological Properties of Antigens and Corresponding Immunological Mechanisms

Lokhandwala et al. highlight that the empirical identification of antigens that are necessary for inducing a protective response, along with the development of an appropriate antigen delivery system to elicit robust cellular and humoral responses, may be a promising strategy for the creation of an effective ASFV vaccine [25]. The selection of antigenic genes is crucial for the development of vector vaccines. Antigens capable of eliciting robust humoral and cellular immune responses have the potential to alleviate the disease or even improve survival, so we have collated the immunological properties of the reported ASFV antigens in Table 1. We also discuss the specific mechanisms related to immune responses elicited by antigens upon their entry into the host. However, it must be emphasized that antigenic proteins can sometimes induce lethal immunity and exacerbate symptoms instead of alleviating them [26]. Theoretically, all viral proteins, if expressed in sufficient amounts, will induce an immune response; therefore, evaluating the protective effectiveness of experimental vaccines through in vivo challenges with ASFV is the most convincing and efficient approach [7].

In Table 1, we list the antigenicity, ability of the antigen to induce an antibody response, and ability to promote specific T cell responses of the major epitopes of ASFV. Additionally, the mechanisms of related virulence genes and immune escape genes are elucidated. Antigenicity does not inherently guarantee protective efficacy; in fact, there are instances where it may negatively correlate with the level of protection offered [7]. Zsak et al. conducted passive transfer experiments in pigs to demonstrate the role of antiviral antibodies in conferring homologous protective immunity against the virulent ASFV strain E75. Notably, 85% of the animals treated with anti-ASFV immunoglobulin survived the infection, whereas the control group exhibited a mortality rate of 100% [27].

**Table 1 ijms-26-00921-t001:** Summary of characteristics of main antigen determinants of ASFV.

Gene Name	Protein	Mechanism	Antigenicity ^1^	Induces Antibody Response ^2^	Induces CTL Response ^3^	Reference
E183L	p54	Participates in viral particle assembly and viral adhesion to host cells. Induces apoptosis in the late stage of infection.	Yes	Yes/Low	Yes	[25,28,29,30,31,32,33]
EP153R	C-type lectin	Participates in the process of blood cell adsorption. Regulates apoptosis and inhibits MHC-1 expression.	-	Yes	Yes	[33,34,35,36]
EP402R	CD2v	Interacts with cellular AP-1 protein and participates in intracellular transport of the virus. Inhibits lymphocyte proliferation.	Yes	Yes	Yes	[25,33,37,38,39,40,41,42]
B119L	9GL	Influences virion maturation and viral growth in macrophages and virulence in swine.	-	Yes	Yes	[30,43,44]
B646L	p72		Yes	Yes	Yes	[25,28,31,45]
B602L	p72 chaperone	Inhibits, causing a decrease in, p72 expression, and inhibits the processing of pp220 and pp62. Its absence severely alters virus assembly.	-	Yes	Yes	[31,33,43,46]
CP204L	p30		Yes	Yes	Yes	[25,28,31,33,47]
CP2475L	pp220		-	Yes	Yes	[45,47,48]
DP71L	NL	Inhibits the activation of the peIF2α-ATF4-CHOP signaling pathway and its mediated apoptosis.	Yes	Yes	Low	[33,49,50]
CP530R	pp62		Yes	Yes	Yes	[21,25,51,52]
MGF360-11L	KP362L	Inhibits the transcription of type I IFN and other cytokines. Increases the survival of infected cells.	-	-	Yes	[33]
MGF505-4R	-		-	Yes	Yes	[17,33]
MGF505-5R	A498R		-	Yes	Yes	[17,33]
F317L	-		Low	Yes	Yes	[33,53]
EP364R	-	Has a nuclease function, inhibiting the IFN signaling pathway of the host’s antiviral innate immune response.	-	-	Yes	[44,54]
E119L	j18L		-	-	Yes	[33,55]
I329L	TLR inhibitor	Inhibits TLR signaling. Inhibits the NF-κB and IRF3 signaling pathways.	Yes	Yes	Low	[33,37,56]

^1^ Antigenicity: could be recognized in sera from pigs who have recovered from ASFV infection. ^2^ Induces antibody response: could induce antibody responses after immunization in vivo. ^3^ Induces CTL response: could induce specific T cell responses after in vivo immunization.

Antibodies play an important role in protective immunity against ASFV, which corresponds to the immunogenicity of the viral proteins. ASFV-neutralizing antibodies have been reported since 1993; however, the potential role of these neutralizing antibodies in immune protection remains controversial [57]. Antibodies targeting the viral proteins p72, p54, and p30 have been demonstrated to neutralize the virus by inhibiting viral binding and internalization [28]. While neutralization is generally absent or poor following natural ASFV infection, Silva et al. demonstrated for the first time in 2022, using extensive data, that there is a high correlation between the presence of virus-neutralizing antibodies and protection [58]. In their study, 95.7% of animals vaccinated with live attenuated vaccines who survived an ASFV attack showed virus-neutralizing activity [58]. The study of Silva et al. reinforces the importance of antibodies in protective immunity. However, it is important to emphasize that the speed and efficiency of neutralizing antibodies can vary by more than 1000-fold between different viruses, or even among different antibodies targeting the same antigenic site on a given virus. In some cases, neutralizing antibodies may exhibit antagonistic effects, and certain in vitro studies have demonstrated that antibodies can upregulate cell infection, leading to the antibody-dependent enhancement of the infection [57]. In addition to this, protective antibodies not only neutralize the virus to impede viral infection, but also function through various mechanisms, such as antibody-dependent cellular cytotoxicity (ADCC), antibody-dependent cellular phagocytosis (ADCP), and complement-dependent cytotoxicity (CDC) [58], as well as conditioning and synergizing with T cell-mediated protective mechanisms [59]. Among these, ADCC has been shown to play an important role in anti-herpesvirus infections, and is effective against porcine pseudorabies virus (PRV) infections. The main effector cells of ADCC activity are polymorphonuclear leukocytes (PMNs), which are more sensitive to lower levels of antibodies than other Fc receptor cells [60]. ADCC can be mediated by natural killer (NK) cells with FcγRIII and monocytes with FcγRII. In the case of NK cells, for example, after the hemagglutinin antibody binds to FcγRIII, NK cells release perforin and granzyme to kill the target cells. In contrast, ADCP involves the activation of FcγRII on the surface of macrophages by antibody-conditioned target cells to induce phagocytosis, which leads to the internalization and degradation of target cells through phagosomal acidification [61]. CDC involves the complement lysing target cells through specific antibodies that bind to corresponding antigens on the cell membrane surface. This binding activates the classical pathway of the complement, ultimately leading to the lysis of target cells via the formation of a membrane attack complex.

Both effective B cell- and T cell-specific responses generally play an important role in the protection afforded by ASFV vaccines [26]. Lokhandwala et al. emphasized that the induction of cytotoxic T lymphocytes (CTLs) may be critical for achieving complete protection [25]. Despite this, few in vitro studies have focused on identifying ASFV proteins that induce cytotoxic T lymphocytes (CTL/CD8+ lymphocytes). Research indicates that, following infection with the ASFV strain OUR/T88/3, the generated antibodies alone are insufficient to protect pigs from subsequent OUR/T88/1 infection. The depletion of CD8+ lymphocytes results in a loss of complete protection against the OUR/T88/1 strain, which reinforces the importance of CTLs in protective immunity against ASFV. Furthermore, pigs that exhibit high antibody titers but low cellular responses following immunization with live attenuated viruses are likely to develop chronic diseases [62]. The interaction between CD8 molecules on CTLs and the class I major histocompatibility complex (MHC-I) is significant, as the latter binds to peptides derived from cytoplasmic pathogens and presents them to the T cell receptor (TCR) [26]. Therefore, CD8 serves as an important marker of CTLs, and the presentation of ASFV antigens by the class I MHC is particularly critical. The immunological mechanisms involved are detailed in Figure 1.

In addition to CTLs, T helper cells (Th/CD4+ T cells) are also major functional cells in cellular immunity, and play an important role in the fight against viral infections [63]. The phenotype of porcine T lymphocytes differs markedly from that of other species in the presence of two populations of T helper cells: CD4+CD8- cells with the classical phenotype of T helper cells, and the distinctive CD4+CD8+ T lymphocytes. The latter co-express CD8α, and pigs are the only species known to contain such large numbers of CD4+CD8+ T lymphocytes in the blood and extra-thymic lymphoid tissues [64]. Following ASFV infection, antigen-presenting cells present immunogenic peptide-MHC class II (pMHCII) to activate CD4+ T cells, which differentiate into CD8+ memory cells and interact with B cells to induce antibody secretion and class switching [65,66].

#### 3.1.2. Antigen Types

Mining potentially valuable protective antigens for vaccine development has been the focus of previous ASFV vaccine research. James et al., through immune protection experiments, screened out several strong immunoreactive proteins—CP204L, F317L, MGF505-4R, MGF360-11L, B602L, EP364R, and E183L—from 44 gene products of the ASFV proteome [33]. Based on published research results, we summarize several classes of antigens or combinations of antigens that are capable of stimulating a strong immune response and of providing protective efficacy.

(1)I329L

The I329L protein is a type I transmembrane protein encoded by the *I329L* gene [67]. It is expressed on the viral envelope and the surface of infected cells during the late stages of ASFV infection, and its gene is relatively conservative [68]. Structurally similar to Toll-like receptor (TIR) family proteins, I329L can specifically bind to and inhibit the activity of TIR domain-containing adaptor-inducing interferon-β (TRIF), a key junction protein in the Toll-like receptor signaling pathway. This inhibition subsequently blocks the activation of interferon regulatory factor 3 (IRF3) and NF-κB [69], thereby preventing the downstream transcription of interferon (IFN). Few studies have focused on the I329L antigen. Until 2023, studies confirmed its antigenicity and ability to induce robust immune responses, leading to its application in indirect enzyme-linked immunosorbent assays (iE-LISAs). The I329L-ELISA was developed for the detection of ASFV antibodies [56], which means that I329L has become a candidate antigen for vaccine research, and is expected to be further used in this area of research.

(2)p30

The p30 protein, encoded by the *CP204L* gene, serves as the internal envelope structure protein of ASFV, which is expressed in the early stage of ASFV infection and mediates the virus as it is invading cells [47,70]. Gómez-Puertas et al. demonstrated, by designing proteinase K isolation experiments, that p30 antibodies inhibited viral internalization by more than 95% in Vero cells and porcine macrophages, suggesting that antibodies that are directed against p30 act by inhibiting viral internalization [28]. Currently, p30 is employed for the serological detection of ASFV [47], which primarily utilizes ELISA and Western blot techniques to identify antibodies, which have demonstrated high accuracy. Furthermore, p30 is recognized as a highly immunogenic protein that is capable of eliciting a robust antibody response during ASFV infection. In the immunity and stimulation tests conducted by James et al., the antibody response to the p30 protein was the highest in the test pigs, and it contained epitopes that stimulated a CTL response, which could cause strong cellular immunity [33].

(3)CD2v

The CD2v protein is encoded by the gene *EP402R*, also known as pEP402R, which is an analog of the surface antigen CD2 of T lymphocytes [68], and the only characteristic viral protein located on the outer envelope of the virus [71]. With a structure that is similar to CD2, CD2v can bind to CD2 ligands on porcine red blood cells [72], facilitating the adhesion of red blood cells to infected cells and enabling the binding of ASFV particles to red blood cells. This process of red blood cell adsorption further promotes the infectious spread of the virus within the host [73]; thus, CD2v is also the hemagglutinin (HA) of ASFV. Studies have demonstrated that, when CD2v is constructed on the baculovirus vector, pigs who are immunized with CD2v generate hemagglutination inhibition and transient infection inhibition antibodies that recognize 75 kDa structural proteins, offering protection against fatal infections [74]. Furthermore, additional experiments have revealed that the inoculation of pigs with the deletion mutant virus BA71 Δ CD2 not only protects against the lethal effects of the parent BA71 strain, but also against the heterologous E75 strain (all genotype I strains). Subsequent studies have indicated that pigs who are immunized with BA71 Δ CD2 exhibit 100% survival against a fatal attack from the Georgia 2007 strain. These results have promoted research on the use of CD2v in the ASFV cross-protection vaccine [75].

(4)p72

The p72 protein, encoded by the *B646L* gene, is the capsid structure protein of ASFV, and is related to the adsorption and invasion of ASFV. p72 is recognized as a key protective antigen, and it has been shown in earlier studies that antibodies against p72 can inhibit the first step of the viral replication cycle, associated with viral attachment [28]. Furthermore, its monoclonal antibody has demonstrated the ability to neutralize virulent ASFV isolates [76]. Research indicates that adenoviral vectors constructed with p72 elicit stronger IgM and IgG responses, along with the secretion of specific IFN-γ, in pigs [25]. As a highly immunogenic genetic target characterized by conserved gene sequences, p72 has been selected for a large-scale evaluation of ASFV’s genetic diversity. Currently, commercial identification tools, such as enzyme-linked immunosorbent assays (ELISAs) based on p72, and colloidal gold strip detection methods utilizing both p72 and its companion protein, the pB602L trimer, have been developed [77].

(5)p54

The p54 protein is the internal envelope protein of ASFV, and is encoded by the *E183L* gene. It is expressed during the early stages of virus replication, plays a role in the adhesion of virus particles to host cells, and can induce apoptosis following viral entry. In addition, antibodies against p54 have been proven to block ASFV adsorption at an early stage, reducing the ability of ASFV to infect cells [28,78]. P54, along with p72 and p30, is regarded as one of the most immunogenic structural proteins. The amino acid sequence of p54 does not overlap with those of other viruses, making it an ASFV-specific protein and a candidate for ASF detection and vaccine development. Stimulation with p54 has been shown to significantly increase serum IgG and mucosally secreted IgA levels, as well as the number of CD4 and CD8 T cells and the expression of IFN-γ, while also enhancing lymphocyte proliferation [79]. Furthermore, its N-terminal domain has been demonstrated to elicit a robust antibody response [80]. Consequently, p54 is extensively utilized in research on and the development of various vaccines, including viral vectors, monoclonal antibodies, and even oral vaccines [79].

(6)pp220

Polyprotein pp220 is a core–shell structural protein encoded by the *CP2475L* gene. During infection, pp220 is processed and cleaved into p150 and pp90 proproteins, with the latter being further cleaved into p34 and precursor pp55 proteins, resulting in the formation of p5, p14, and p37. These decomposed proteins are involved in the nuclear and cytoplasmic transport of viral DNA, and provide protection against the inflammasome, which is essential for the replication of ASFV in host cells [81]. Notably, pp220 contains multiple epitopes that are 100% conserved across different ASFV genotypes and can bind to various SLA-I alleles. Although few studies on the immune effects of pp220 during the early stages have been conducted, a study by Michelle et al. in 2022 demonstrated that immunizing pigs with a fusion of pp220 and an adenovirus vector induced significant IgG production, as well as the activation of IFN-γ-secreting cells and cytotoxic T lymphocyte responses [48]. This suggests that pp220 may serve as a promising candidate target for future research on ASFV vaccines.

(7)MGF360 and MGF505

MGF360 and MGF505 are ASFV polygenic family proteins located in variable regions at the end of the genome. There are five types of MGF within this polygenic family, each exhibiting different complementarities across various ASFV strains [17]. Among them, MGF360 and MGF505 have been identified as significant virulence factors of the virus, influencing the host range of ASFV infection [82]. These proteins can inhibit the type I interferon (IFN) response, thereby enhancing both the efficiency and quantity of viral proliferation by prolonging the survival time of the host [83]. Research has demonstrated that MGF360-12L plays a crucial role in enabling effective viral replication and attenuating porcine infection in macrophages. Additionally, MGF505-11R is capable of inhibiting the cGAS-STING signaling pathway, while MGF505-7R can impede cGAS-STING signal transduction through the autophagic degradation of STING. It can also inhibit the interleukin-1β (IL-1β), type I IFN, and JAK-STAT signaling pathways [84]. MGF505-4R inhibits the IFN-a/b induction pathway, suggesting a novel function for this viral protein in immune evasion [85]. Despite the progress made in functional studies of individual proteins from these two families, a scarcity of protective induction tests for single proteins remains, which has limited the application of the MGF family in vaccine antigens. Currently, only MGF505-4R and MGF360-11L have been identified as capable of inducing strong immune responses [33].

#### 3.1.3. Difficulties and Outlook

Multiple ASFV antigens, highlighted in Table 1, have been shown to elicit robust immune responses in pigs. However, it must be acknowledged that the current research indicates that several single-antigen and even multi-antigen combined viral vector vaccines may not offer comprehensive immune protection. Additionally, the effectiveness of certain recognized strong immunogenic antigens can vary depending on the vector used. In some instances, the disease may even worsen. Although p72 and p54 have been shown to elicit specific antibody responses, Gotley et al. were unable to detect antibody responses to p72 in rAd-primed and MVA-boosted pigs, and found that the antibody responses to p54 in rAd-primed and -boosted pigs were also not ideal. The reason for this may be the selection of an inappropriate carrier combination for primary immunization and booster administration, indicating that variations in carriers can influence the effectiveness of the antigen [86]. Moreover, pigs immunized by Gomez-Puertas et al. with recombinant baculovirus vaccines using p54 and p32 were not protected from fatal infections [87]. Similarly, Neilan et al. utilized the baculovirus recombinant proteins p30, p54, p22, and p72, and found that although vaccines using these proteins were able to generate antibodies, they ultimately failed to protect the pigs from toxic challenges [88]. These experimental results, which demonstrate incomplete protection, further highlight that one of the primary challenges in the development of ASFV vector vaccines is the insufficient exploration of viral genome antigen genes. This also encourages us to expand our focus beyond a limited selection of classic antigens. Exploring novel antigen combinations may enhance vaccine efficacy.

However, the research results in this area are not entirely negative; in fact, some studies utilizing mixtures of multi-antigen recombinant viral vectors have yielded promising outcomes, as detailed in Table 2. Notably, Gotley et al. successfully recombined the eight antigen genes—B646L, CP204L, B602L, E183L, EP153R, E199L, MGF505-5R, and F317L—into eight recombinant adenoviral vectors, and for the first time, immunization was carried out using a mixture of recombinant adenovirus/poxvirus vectors. This approach has been shown to typically express high levels of ASFV-specific IFNγ-secreting cells, induce a specific immune response in pigs, and provide 100% protection against fatal disease following exposure to typically lethal doses of virulent ASFV [86]. Lokhandwala S et al. inserted B646L, CP204L, CP2475L (p37, p150-I, p150-II), CP530R, and E183L genes into seven recombinant adenoviral vectors and administered them in a mixture to study their protective effect. The results showed that the mixture with ZTS-01 as the adjuvant provided 55% protection, and five out of nine pigs survived a lethal dose of Georgia 2007/1, whereas only one out of five pigs survived in the control group [21].

### 3.2. The Introduction of Exogenous Cytokines

In recent years, cytokines have emerged as important diagnostic, prognostic, and therapeutic agents for human and animal diseases [94], playing a crucial role in combating viral and bacterial infections, maintaining immune balance, and regulating cell growth. Previous studies have shown that cytokines, as a novel immunostimulant [95], are mainly used in viral vaccines, and have significant efficacy in the treatment of swine diseases [96]. Porcine interferon (PoIFN) inhibits a variety of viruses, including ASFV, PRV, and porcine reproductive and respiratory syndrome virus (PRRSV). Furthermore, porcine interleukin-21 (pIL-21) has been shown to enhance the immune response against infection with foot-and-mouth disease virus (FMDV) [97]. For decades, researchers have been investigating the possibility of bypassing the production step and providing chemokines or cytokines directly in vaccines [98]. The concept of inserting exogenous cytokines into viral, bacterial, and yeast vectors for expression would also solve protein storage and transportation problems. The following section will specifically address several types of cytokines whose use is supported by data, and which can synergize with ASFV vectors to provide protection, as well as those that are considered to have potential for application in the context of the research focus of our laboratory.

(1)pIL-21

Porcine cytokines, such as PoIFN-α, PoIFN-γ, and pIL-21, exhibit significant inhibitory effects on ASFV. pIL-21 is a crucial cytokine involved in antiviral humoral immunity. IL-21 enhances the expression of CD86 in mouse B cells, thus providing potent T cell co-stimulation [99]. Since IL-21 deficiency leads to memory B cell defects [100], Chen et al. suggest that IL-21 is also a potential vaccine adjuvant. By constructing a recombinant *Lactobacillus plantarum* expressing the ASFV p54 and pIL-21 fusion protein, Chen et al. demonstrated that the expressed fusion proteins elicited specific humoral immunity, as well as mucosal and T cell-mediated immune responses, in mice. Furthermore, the presence of pIL-21 was found to facilitate the proliferation and differentiation of immune effector cells, including T cells, B cells, and natural killer cells [79].

(2)PoIFN-α and PoIFN-γ

PoIFN-α and PoIFN-γ are classified as type I and type II PoIFN, respectively, and they play a crucial role in innate immunity against various viral infections. PoIFN-γ can upregulate macrophage functions, including antigen processing and presentation, as well as the production of cytokines and chemokines. It enhances the antibacterial activity and pro-inflammatory response of macrophages, and promotes inflammation by recruiting lymphocytes, such as NK cells [26]. Additionally, it can reduce ASFV replication in porcine monocytes and alveolar macrophages, inhibiting late-stage protein expression (specifically p220 and p72) rather than early protein expression (such as p27) [101]. Studies have demonstrated that PoIFN-α and PoIFN-γ can inhibit the replication of ASFV, both in vitro and in vivo [102], while also inducing the production of interferon-stimulated genes and major histocompatibility complex molecules [103]. Furthermore, in combination with several production methods of PoIFN, when compared with its traditional expression in *E. coli*, PoIFN has been successfully integrated into viral vectors such as adenovirus and baculovirus for expression, yielding products with the required activity and fewer by-products [104]. Additionally, due to the ability of viral vectors to penetrate cells, PoIFN can be delivered directly to infected cells, thereby enhancing both the persistence and efficacy of the treatment. This approach offers a novel strategy for utilizing ASFV vector vaccines in conjunction with cytokines, presenting a distinct advantage of viral vectors over other types of vaccines.

(3)IL-1β

IL-1 exists in three forms: IL-1α, IL-1β, and IL-1Ra. Of these, IL-1α and IL-1β exhibit strong pro-inflammatory effects. IL-1β acts as an endogenous pyrogen, produced and released during the early stages of the immune response to infection, pathology, and stress. During inflammation, IL-1β stimulates the production of acute-phase proteins in the liver and other cells. Additionally, it serves as a chemoattractant for granulocytes, enhances the expansion and differentiation of CD4+ T cells, and increases the expression of cell adhesion molecules on leukocytes and endothelial cells [105]. Due to its pro-apoptotic effect, Franzoni et al. speculated that the release of IL-1β following BA71V infection in pigs may contribute to the early apoptosis of monocytes/macrophages, potentially limiting viral replication and alleviating pro-inflammatory dysregulation in vivo [101].

(4)IL-18

IL-18 is a member of the IL-1 family and serves as an inducer of IFN-γ production. It works synergistically with IL-12 to activate T and NK cells, and with IL-23 to enhance other T cell responses, such as Th17, or to enhance the Th2 response in the absence of IL-12, IL-15, or IL-23 [106]. Unlike IL-1β, IL-18 is not a pyrogen, and even attenuates the febrile response induced by pyrogenic IL-1β [107]; its release may be associated with the development of TH1 cell responses and protection against ASFV.

A growing number of studies in various fields are considering the exogenous expression of cytokines. It has been shown that increasing IL-21 production or using exogenous methods of increasing cellular IL-21 production can limit the extent of initial HIV-1 infection [108]. The use of exogenous cytokines holds promise as a means of enhancing protection.

### 3.3. Types of ASFV Vector Vaccines

The entry of exogenous target DNA fragments into the host cell for efficient expression cannot be achieved without the help of delivery systems, which protect the DNA from invading endosomes and lysosomes that cross the nuclear membrane and other biological barriers to reach the nucleus for transcription. More and more delivery systems are being developed to enhance immunogenicity and the overall immune response [109], such as nanoparticle complexes and viral vectors. Nanoparticles are able to encapsulate DNA fragments through electrostatic interactions or chemical bonding, ensuring their stable circulation in the bloodstream for long periods of time and facilitating target tissue aggregation [110]; nanoparticles are also able to interact with immune receptors and proteins, thereby disrupting signaling cascades [111]. However, nanoparticles suffer from nucleation difficulties, low transfection efficiency, and limited transfection capacity [112], problems that are addressed by viral vector delivery systems. In contrast to non-live vaccines, which can only rely exclusively on cross-presentation to induce immunity, viral vector vaccines utilize the host’s translational system to express genes and gain direct access to the host’s MHC class I peptide-loading machinery by direct presentation [113]. Specifically, viral vectors exhibit high infection efficiencies, and have an inherent mechanism for entering cells and overcoming endosomal constraints to facilitate DNA delivery to the cytoplasm. Viral vectors have well-organized structures and diverse nuclear localization signaling proteins (NLSs) that efficiently recognize nuclear transporter proteins and subsequently deliver DNA to the nucleus for expression [112]. Viral vectors also act as adjuvants, inducing a transcriptional program that activates infected antigen-presenting cells, thereby stimulating the adaptive immune system to produce potent antibodies and a Th1-biased cellular response, including CD8+ T cells [114]. In addition to viral vectors, bacterial and yeast vectors are also described below.

#### 3.3.1. Viral Vectors

(1)The adenovirus vectors

Adenoviruses are unenveloped, linear double-stranded DNA viruses that are encapsulated by icosahedral capsids. Their hosts include a variety of vertebrates, such as humans, mice, pigs, dogs, and non-human primates. The genome length of adenoviruses typically ranges between 35 and 36 kb, with reverse repeat sequences at both ends. Adenoviral vectors have emerged as widely utilized viral vectors, serving as mature delivery systems for gene function analysis and therapeutic research. With a large transgene capacity, high transduction efficiency, and robust thermal stability, these vectors can also induce innate immunity and immune responses, due to the expression of exogenously inserted gene proteins.

As shown in Figure 2a, when exogenous genes are expressed using adenoviral vectors, the E1A and E3 genes are typically deleted from the viral genome to facilitate transgene insertion. The E1A gene, located within the left inverted terminal repeat (ITR) sequence, is essential for viral replication, as the expression of the E1A protein activates the transcription of the virus’s early genes. The deletion of the E1A gene renders the Ad vector incapable of replication; however, the E3 gene does not influence viral replication. The modified Ad vector can proliferate in HEK293 cells that stably express the Ad5 E1 protein [115].

The adenovirus vector vaccine for ASFV has been extensively studied, with successive results demonstrating its feasibility and potential. The C adenoviruses of serotypes 2 and 5 adsorb to host cells through a high-affinity interaction between the knobbed structural domains of their fibers and the cell surface receptor, the Coxsackie/adenovirus receptor (CAR) [116,117]. Subsequently, cell-surface integrins interact with specific motifs located in the viral capsid to initiate viral internalization via receptor-mediated endocytosis [118,119]. Upon entry, the viral particle undergoes a complex catabolic process that produces the viral capsid, and is transported along microtubules to the nuclear pore complex, where the viral genome is imported into the nucleus [120,121]. Unlike retroviral infection, which requires host mitosis, adenoviral infection is not dependent on the cell cycle state, and rarely involves integration of viral genes into the host genome [122,123]. These features enable a wide range of infected cell types, and reduce the risk of chromosomal mutations. In addition to the aforementioned advantages, research indicates that adenovirus vector vaccines can elicit more robust cytotoxic T lymphocyte (CTL) responses compared with the vaccinia virus, plasmid DNA, or a combination of both [43]. Utilizing primary adenovirus immunization, along with an enhanced delivery system based on the improved vaccinia Ankara, Goatley et al. employed eight recombinant adenovirus vectors to protect pigs from fatal disease caused by genotype I ASFV strains [86]. The recombinant vector, constructed from adenovirus vector Ad5 and an antigen gene, significantly alleviated the clinical symptoms and viremia in pigs who were infected by the virulent strain OUR T88/1 [51].

Recent studies have demonstrated that the adenovirus vector ASFV vaccine can effectively protect pigs from ASFV infection. Liu et al. utilized effectively replicated Ad2 in pigs to construct the recombinant ASFV vectors Ad2-p30, Ad2-p54, Ad2-p30-p54, Ad2-CD2v, and Ad2-p72-p72c. Their findings indicated that this Ad2-ASFV cocktail-based immunization can successfully induce systemic IgG antibodies and lung IgA antibodies against ASFV antigens when administered through a combination of intramuscular and intranasal routes [124]. Following the experiments, the combination of intramuscular and intranasal immunization exhibited favorable immunogenicity and safety, thereby providing a promising immune strategy for subsequent research.

(2)The herpesvirus vectors

Herpesvirus is an enveloped DNA virus that is categorized into three subfamilies: α, β, and γ. To date, more than 100 types of herpesviruses have been identified. Pseudorabies virus, a member of the *Herpesviridae* family, is extensively utilized in research involving ASFV vectors.

PRV is the causative agent of porcine pseudorabies, and is classified as a double-stranded DNA virus with a genome size of 143 kb [125]. PRV contains more than 70 ORFs that encode between 70 and 100 proteins; however, only about 50 distinct proteins are present in mature viruses, and many of these genes have minimal relevance to viral replication. Among them, the gC gene encodes the major glycoprotein that mediates the attachment of PRV to target cells [126,127], whereas the glycoproteins encoded by gB, gC, and gD induce protective immune responses [128,129]. As shown in Figure 2b, the TK, gE, gG, and gI loci are commonly utilized for the insertion of exogenous genes, and their deletion does not impede viral replication, resulting in a reduced virulence phenotype [130]. In particular, the double knockout of the TK and gE genes eliminates the depletion of T cells, B cells, and monocytes/macrophages in the blood caused by wild-type viral infections; reduces granulocyte proliferation to eliminate granulocyte immunosuppression of T cells; and enhances the immune system’s ability to fight PRV infection [78]. This characteristic makes these genes suitable for the development of recombinant vectors, such as gE^−^/gI^−^/TK^−^ PRV (HeN1) [131], PRV-TK^−^/gE^−^/gI^−^ (Fa) [132], and PRV-ΔgE/ΔgI/ΔTK(CD2v) [133]. The advantages of the PRV expression vector system are similar to those of other viral expression vectors, in that it is safe, has a large capacity for exogenous genes, has a low production cost, is relatively simple to use, and can be safely used for effective expression of exogenous genes for the development of multivalent genetically engineered live vaccines [44].

Feng et al. developed a recombinant vaccine, PRV-ΔgE/ΔgI/ΔTK-(CD2v), which expresses the ASFV CD2v gene using PRV as a vector. They verified its safety and immunogenicity in mice. The PRV-ΔgE/ΔgI/ΔTK-(CD2v) recombinant strain demonstrates strong immunogenicity, safety, and efficacy, and can produce anti-CD2v-specific antibodies, activating a specific cellular immune response [133].

Deng et al. utilized CRISPR/Cas9 gene editing technology to construct a live attenuated recombinant pseudorabies virus vector, rPRVXJ-EGFP/B602L/B646L, which expresses the p72 and pB602L proteins. This vector was shown to elicit specific humoral and cellular immune responses against both p72 and pB602L, providing complete protection (100%) against PRV strains (PRV-XJ) [90].

In a recent study, Geng et al. constructed two recombinant pseudorabies virus vectors, rGXGG-2016-ΔgI/ΔgE-EP364R and rGXGG-2016-ΔgI/ΔgE-B119L, expressing the ASFV EP364R and B119L proteins, respectively, and verified that both vectors were immunogenic in mice and induced the production of specific antibodies against expressed ASFV proteins, while providing protection against lethal PRV attacks. This work also highlights the potential value of EP364R and B119L in ASFV vaccine development (as previously summarized in Table 1) for the construction of bivalent ASFV and PRV vaccines [44]. A series of studies have demonstrated that the recombinant PRV strain exhibits strong immunogenicity and is capable of inducing robust humoral and cellular immune responses.

(3)The alphavirus vectors

Alphavirus is an enveloped, single-stranded, positive-sense RNA virus. More than 30 members have been identified, including the Sindbis virus (SINV), Venezuelan equine encephalitis virus (VEEV), Semliki forest virus (SFV), Chikungunya virus (CHIKV), and Eastern equine encephalitis virus (EEEV). The alphavirus vector carries the alphavirus RNA polymerase gene, enabling it to simulate viral replication with a low mutation rate. This vector can stably and efficiently express exogenous genes; thus, the alphavirus vector vaccine exhibits superior duration and immune response compared with other vaccines. Additionally, the relatively small RNA genome of the alphavirus allows it to be easily manipulated when it is replicated beforehand as a cDNA. Furthermore, the alphavirus can stimulate strong IFN responses [134]. Murgia et al. utilized the alphavirus replicon particle (RP) system as a carrier to express exogenous ASFV antigens, specifically p30 (RP-30), p54 (RP-54), and pHA-72 (RP-sHA-p72). Their findings confirmed the immunogenicity of these antigens, ranked from highest to lowest as RP-30, RP-54, and RP-sHA-p72 [135].

(4)The poxvirus vectors

Poxvirus is a double-stranded DNA virus, and research on this virus has been ongoing for over 100 years. Currently, the primary poxvirus vectors include vaccinia virus, cowpox virus, fowlpox virus, and swine pox virus. Recombinant poxvirus vectors can accommodate large fragments of exogenous DNA (up to 25 kb), facilitating the development of multivalent vaccines through the insertion and expression of multiple genes [136]. These vectors have been demonstrated to induce lasting immunity while activating both the humoral and cellular immune responses. Furthermore, the structural similarities between ASFV and poxvirus suggest that recombinant ASFV proteins can be effectively expressed within poxvirus vectors.

Modified vaccinia virus Ankara (MVA) is recognized as one of the most advanced poxvirus vaccine vectors. It is an extremely safe non-replicating strain, and can be studied under Biosafety Level 1 (BSL-1) conditions. In addition, it exhibits large packaging capacity for exogenous DNA, tight control of recombinant gene expression, vaccine immunogenicity and efficacy, and ease of vector and vaccine production [137]. Numerous experiments have demonstrated the immunogenicity of the ASFV antigen that MVA expresses [86,138]. By constructing several recombinant MVA vectors, Lopera-Madrid et al. showed that high levels of p30 mRNA and specific anti-P30 antibodies were induced in mice by the natural poxvirus PrMVA13.5L promoter [139]. Additionally, this study highlighted that the synthetic PrS5E promoter and the SE/L promoter, which is linked to the secretion signal, yielded lower mRNA levels and antibody responses. This finding underscores the importance of selecting an appropriate promoter, which is as critical as choosing the right antigen.

(5)Newcastle Disease Virus vectors

Newcastle disease virus (NDV) is a polymorphic enveloped virus that belongs to the family *Paramyxoviridae*, and is a well-characterized member of the avian paramyxovirus serotype [140]. The genome of NDV is an unsegmented, negative-sense, single-stranded RNA with six genes coding for separate transcriptional units: nucleocapsids (N), matrix proteins (M), phosphoproteins (P), fusion proteins (F), hemagglutinin-essential neuraminidase proteins (HN), and large polymerase proteins (L). The P gene encodes three proteins through the phenomenon of RNA editing. The remaining five genes encode a single protein. Each gene on the NDV genome is marked by the presence of gene-start and gene-end signal sequences. Transcriptional reinitiation of the intergenic region [141] results in the genes close to the 3’-end showing a transcriptional gradient of higher levels of expression [142].

It has been shown that NDV is a promising vaccine vector in the development of vaccines that can be delivered to the respiratory tract [143]. Intranasally delivered NDV vectors induce a strong cellular immune response against vaccine antigens [144]. NDV replication in mammalian cells induces strong expression of interferon-α/β, which acts as a natural adjuvant to promote the maturation of antigen-presenting dendritic cells and the establishment of a robust immune response [145,146]. Additionally, the modular NDV genome allows for the direct insertion of exogenous transcriptional units, and there does not appear to be a strict limit on the length of genomic RNA that can be packaged in the virosome [142]. These features give NDV an important advantage over other viral vectors. Chen et al. constructed a recombinant NDV vaccine, rNDV/p72, expressing the ASFV p72 protein, and performed immunoassays in mice. The results showed that rNDV/p72 successfully induced a high titer of p72-specific IgG antibody, and promoted the proliferation of T cells, as well as the secretion of IFN-γ and IL-4 [147].

#### 3.3.2. Bacterial Vectors

Bacterial cells are an attractive alternative as vectors to deliver recombinant antigens for the development of new vaccines. An important advantage of live bacterial vaccines is their potential for exploiting the immune response of the mucosal system. The mucosal vaccination strategy is generally regarded as having fewer side effects, a lower production cost, and easier implementation. Recombinant strains of lactic acid bacteria and the attenuated *Salmonella* bacterium have been constructed for the delivery of ASFV antigens.

(1)Lactic acid bacterial vectors

Lactic acid bacteria (LAB) are not only generally regarded as safe (GRAS), but are also able to act as adjuvants to stimulate the host’s immune system, due to their probiotic properties and immunomodulatory capacity [148,149,150,151]. This renders them highly promising as safe and efficacious vector candidates for the delivery of ASFV antigens. Various genetic engineering tools have been developed for *Lactococcus lactis* to efficiently express antigenic and therapeutic molecules in different cellular localizations (i.e., cytoplasm, cell wall, or extracellular media) [152]. Zhang et al. constructed a recombinant *Lactococcus lactis* strain expressing the ASFV p30, p54, and p72 proteins. Rabbits that were orally immunized with this recombinant *Lactococcus lactis* strain produced high levels of serum-specific IgG antibodies, small intestinal mucosal sIgA antibodies, and cytokines (IL-4 and INF-γ). This suggested that the recombinant *Lactococcus lactis* strain expressing p30, p54, and p72 proteins promoted mucosal, humoral, and cellular immunity against ASFV [91].

*Lactobacillus plantarum* has been used to express the porcine epidemic diarrhea virus S gene of the S-DCpep fusion protein against infectious gastroenteritis virus [153]. Chen et al. constructed the recombinant *Lactobacillus plantarum* strain expressing the ASFV p54 and porcine IL-21 fusion protein. Feeding mice with this recombinant *Lactobacillus plantarum* strain significantly elevated the levels of serum IgG and mucosal IgA secretion, increased the counts of CD4+ and CD8+ T cells, and enhanced the expression of IFN-γ in both CD4+ and CD8+ T cells of these mice. Lymphocyte proliferation was also observed following stimulation with the recombinant *Lactobacillus plantarum* strain [79]. Huang et al. used *Lactobacillus plantarum* to express the ASFV p14.5 antigenic protein. Improvements in the differentiation and maturation of T lymphocytes, B lymphocytes, and dendritic cells (DCs) were observed following the oral administration of this recombinant *Lactobacillus plantarum* strain to specific pathogen-free (SPF) mice [92].

Although rabbits and mice are not the natural hosts for ASFV, the above studies provide viable models for evaluating immunogenicity.

(2)The *Salmonella typhimurium* vector

Kiran et al. chose an attenuated *Salmonella typhimurium* strain JOL912 (*ΔlonΔcpxRΔasd*) [154,155] to express the ASFV antigenic genes *B119L*, *EP402R*, *EP153R*, *O61*, *E183L*, and *B464L.* When the recombinant *S. typhimurium* strain was administered intramuscularly in pigs, significant increases in the levels of helper T cells, cytotoxic T cells, NK cells, and immunoglobulin (IgG, IgA, and IgM) were observed [65].

#### 3.3.3. Yeast Vectors

*Saccharomyces cerevisiae* has been certified as generally regarded as safe (GRAS), and is widely used as a vector for the expression of infectious disease or cancer antigens [156,157,158]. The advantages of *Saccharomyces cerevisiae* vectors include rapid growth, ease of replication and isolation of mutants, and a well-defined genetic system [159]. Shuo et al. constructed recombinant *S. cerevisiae* strains expressing the ASFV antigens KP177R, E183L, E199L, CP204L, E248R, EP402R, B602L, and B646L on the cell surface. After oral administration of this yeast recombinant strain in mice, ASFV-specific immune responses, including serum IgG and mucosal secretory IgA, were elicited. CD8+ T cells were activated, and a balanced immune response dominated by Th1 cells was stimulated [93].

## 4. Conclusions

There is an urgent need for an effective vaccine to control African swine fever. Vector vaccines represent a promising approach to overcoming the challenges associated with vaccine development. Viral vectors, including adenovirus, alphavirus, pseudorabies virus, cowpox virus, and Newcastle disease virus, have demonstrated efficacy in eliciting robust antigen-specific immune responses against ASFV. The immune responses elicited by bacterial and yeast vectors during the delivery of ASFV antigens have also been assessed. However, studies evaluating the protective efficacy of some of these vaccines in swine, the natural host of ASFV, have yet to be conducted.

The limited amount of research on the immunogenicity of ASFV antigen proteins and the induction of specific T cell responses has constrained further optimization of these vector vaccines. Numerous studies have revealed the effectiveness and safety of cytokines in the treatment of ASFV. Therefore, integrating newly reported effective antigens, new combinations of antigens, and combinations of antigens and cytokines into appropriate vectors is of great significance for future research.

## Figures and Tables

**Figure 1 ijms-26-00921-f001:**
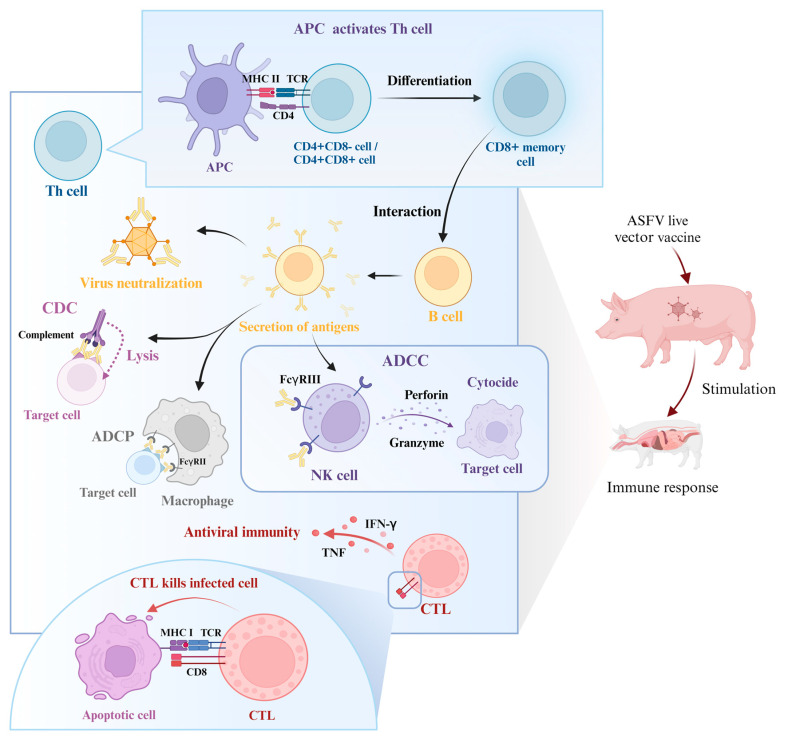
Mechanism of host immune response induced by ASFV vector vaccine. APC, antigen-presenting cell; Th cell, T helper cell; NK cell, natural killer cell; ADCC, antibody-dependent cellular cytotoxicity; ADCP, antibody-dependent cellular phagocytosis; CDC, complement-dependent cytotoxicity; CTL, cytotoxic T lymphocyte; IFN-γ, interferon-gamma; TNF, tumor necrosis factor; MHC I, class I major histocompatibility complex; MHC II, class II major histocompatibility complex; TCR, T cell receptor (created with BioRender.com).

**Figure 2 ijms-26-00921-f002:**
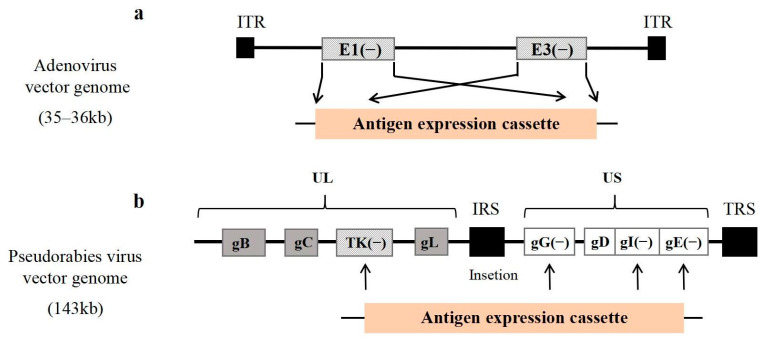
Viral vectors used for ASFV vaccine development. The genomic loci used for the insertion of exogenous antigen cassettes are indicated. (**a**) The adenovirus vector has inverted terminal repeats (ITRs) at both ends of the adenovirus genome, which are often used to replace the exogenous fragment using the E1 and E3 sites. (**b**) The Pseudorabies virus vector has a unique long region (UL) and a unique short region (US) in the Pseudorabies virus genome, which are indicated by gray and white boxes in the figure, respectively. There are also internal and terminal repeat sequences (IRSs and TRSs).

**Table 2 ijms-26-00921-t002:** Summary of protective efficacy of ASFV multi-antigen vector vaccines.

Antigen Genes/ Proteins	Vector Types	Immunization Procedures		Virus Strains and Virulence Measurements	Protective Effect	References
		Prime	Boost			
*B646L*, *CP204L*, *B602L*, *E183L*, *EP153R*, *E199L*, *F317L*, and *MGF505-5R*	Adenovirus and poxvirus	Mixture of eight Adv vectors	Mixture of one Adv vector and seven MVA vectors	104 HAD50 OUR/T88/1	100% protection (six/six pigs)	[86]
*B646L*, *CP204L*, *CP2475L (p37*, *p150-I*, *p150-II)*, *CP530R*, and *E183L*	Adenovirus	Mixture of seven Adv vectors	Mixture of seven Adv vectors	103 TCID50 Georgia 2007/1	55.5% protection (five/nine pigs)	[21]
p30, p54, and psHA	Baculovirus	BacMam-sHAPQ	BacMam-sHAPQ	102 HAU50 E75	66.6% protection (four/six pigs)	[89]
*B646L* and *B602L*	Pseudorabies virus	rPRVXJ-EGFP/B602L/ B646L	rPRVXJ-EGFP/B602L/ B646L	106 TCID50 PRV-XJ	Ten/ten mice survived	[90]
p30, p54, and p72	*Lactococcus lactis*	MG1363/pMG36e-ASFV	MG1363/pMG36e-ASFV	-	-	[91]
p54 and pIL-21	*Lactobacillus plantarum*	NC8pSIP409-pgsA’-p54-pIL-21	NC8pSIP409-pgsA’-p54-pIL-21	-	-	[79]
p14.5	*Lactobacillus plantarum*	NC8-pLP-S-p14.5	NC8-pLP-S-p14.5	-	-	[92]
*B119L*, *EP402R*, *EP153R*, *O61*, *E183L*, and *B464L*	Attenuated *Salmonella typhimurium*	rSal-ASFV	rSal-ASFV	-	-	[65]
*KP177R*, *E183L*, *E199L*, *CP204L*, *E248R*, *EP402R*, *B602L*, and *B646L*	*Saccharomyces cerevisiae*	EBY100/pTy1E −ASFV	EBY100/pTy1E −ASFV	-	-	[93]

The experiments listed in Table 2 are at the trial stage.

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
