# Peer review of "Progress in African Swine Fever Vector Vaccine Development"

_ijms, 2025, doi:10.3390/ijms26030921_

Round 1
Reviewer 1 Report (New Reviewer)
Comments and Suggestions for Authors
dear authors,
I began pointing out one by one all the mistakes and gaps (in my opinion) that I detected in your manuscript. After finishing the introduction section I decided to relax myself a little bit and to not return to you a whole re-written manuscipt. I am a reviwers, not one of the authors. Do please find all my commentar, suggestions, questions, etc. in the attached pdf. Do please feel free to follow them or not.

dear authors,
I am not a native-English speaker, as you will probably notice. But there are som English usage mistakes that even I could detect. I pointed some of them in the attached pdf. I feel that you should give the present paper to a native-English speaker for style-correction before sending it again to the journal.
Author Response
Please see the attachment.

Reviewer 2 Report (New Reviewer)
Comments and Suggestions for Authors
The manuscript "Progress in African Swine Fever Live Vector Vaccine Development" by Yang et al., provides a comprehensive overview of ASFV live vector vaccine. It highlights various viral, bacterial, and yeast-based vectors, ASFV antigens, cytokine integration, and novel combinations to enhance vaccine protection. Overall, this review serves as a valuable resource for researchers in the field of swine vaccination against ASFV.
Some comments are provided below.
1- The genus and family names (e.g., Asfarviridae, Asfivirus) should be italicized for accuracy and consistency.​
2-The authors should clarify why cytokines were chosen over other novel immunostimulants. This would strengthen the rationale for their inclusion in vaccine strategies​.
3-The manuscript should include Newcastle disease virus (NDV) as a promising candidate among viral vectors. Recent research has highlighted NDV's safety profile and robust immunogenicity, making it a potential vector for ASFV vaccine development.
4-While the manuscript provides a comprehensive review of ASFV live vector vaccines, it lacks a detailed discussion on the critical role of delivery mechanisms in determining vaccine efficacy. Addressing this gap would deepen the analysis and offer practical insights for optimizing ASFV vaccine strategies.
Author Response
Please see the attachment.

Reviewer 3 Report (New Reviewer)
Comments and Suggestions for Authors
The authors have implemented properly the scientific backgroud, as suggeste in the last versione.
I just recommend to add another citation regarding p30 protein (see Liberti et al) in the paragraph "p30", line 456 and in table 1, respctively.
Author Response
Comment 1: I just recommend to add another citation regarding p30 protein (see Liberti et al) in the paragraph "p30", line 456 and in table 1, respectively.
Response 1: Thank you very much for your suggestions, we have added the reference in Table 1 and the p30 paragraph (lines 252 and 256), as you suggested. All changes to the revised text are highlighted in red.
We hope that our manuscript is now acceptable for publication in International Journal of Molecular Sciences. If there are any other issues to be addressed, please let me know and we will comply.
Best regards,
Hailong Wang
This manuscript is a resubmission of an earlier submission. The following is a list of the peer review reports and author responses from that submission.
Round 1
Reviewer 1 Report
Comments and Suggestions for Authors
The topic is interesting and should be reviewed in a manner enlightening the reader about the difficulties of making a viral vector vaccine. The articles seems to overlook the fact that despite all the work, there is currently no viral vector vaccine that is sufficiently protective to merit further development. Even Adenovirus vectors that have used a multitude of antigens failed. The review neither mention these problems nor discusses the reasons for them. In the review, several antigens are mentioned as providing protection with often one manuscript (ofter older) cited. I miss pointing on the observation many of these antigens used in viral vectors failed to protect and some even caused enhanced disease. With this, I also miss a discussion on the difficulties in the field. Overall, the critical problems and contradictions in the field of ASF vaccinology are not discussed in sufficient details and not all critical works related to the individual antigens are cited. When it comes to immunology, the review remains superficial only mentioning antibodies and T cells but nothing about their potential functions.
I think that such a review must discuss in more detail the conflicting role of antibodies. I would like to give an example. p72 is indeed very immunogenic and older studies pointed on a role of antibodies against this protein in neutralization and protection. However, other studies using vectors contradict this. Furthermore, ASFV is enveloped and p72 is a capsid protein, so how can these antibodies work? How would antibodies work? All this aspects are ignored in the review and this applies also to other proteins. There is no systemic assessment of the listed protein that enables a conclusion including them as vaccine antigens or not. The immunogenicity of a viral protein alone does not make it a target for vaccine. I am sure the authors are very well aware of this but looking at Table 1 and some of the text can give the impression that only immunogenic proteins would represent valuable targets. This may not necessary be the case.
I also identified a number of wrong sentences and misleading statements, wrong references. I stoped listing them after line 202 because they exceed what is acceptable to me as a reviewer. Writing a review must be done with great care and giving credit to the correct papers is absolutely essential. The problem is that many authors are "lazy" like to cite reviews. So the review must fulfil very high standards.
Comments on the Quality of English Languageok
Reviewer 2 Report
Comments and Suggestions for Authors
The manuscript comprehensively reviews the current progress in developing viral vector vaccines for African swine fever virus (ASFV), a highly lethal virus affecting the swine industry worldwide. The paper focuses on the structure of the virus, antigen selection, different types of viral vectors, and the potential role of cytokine insertions in enhancing vaccine efficacy. Given the global impact of ASF and the lack of an effective commercial vaccine, the review highlights the promising role viral vector vaccines may play in mitigating the disease.
Major Comments:
01. The manuscript offers a broad and detailed exploration of the topic but could benefit from tighter structuring. To improve readability, the discussion on viral vector types, antigen selection, and cytokine inclusion was more clearly segmented, with separate subheadings to highlight key themes and concepts.
02. The authors summarize key advancements in ASFV vaccine research; more in-depth analysis is needed, especially regarding why certain vector types (Adenoviruses, Vaccinia virus, etc.) are preferred over others in specific experimental settings. Furthermore, a critical comparison of efficacy data for Each viral vector platform is necessary to emphasize the advantages and disadvantages.
03. The authors briefly discuss the potential of cytokine insertion to enhance immune responses. This aspect, while promising, is underexplored in the review. A deeper discussion of the current experimental evidence on cytokine insertion's effects on immune modulation would strengthen the manuscript.
04. Several vaccines are mentioned in the review without detailed follow-up on their status (Ex: trials or field tests). Including updated information on their development stages, clinical trials, or practical applications would enhance the relevance of the paper to current research trends.
05. I strongly recommend that the authors introduce a few figures to illustrate the information explained. It would enhance the readability of the manuscript.
Minor Comments:
01. The introduction could benefit from a more concise explanation of why ASFV has resisted vaccine development so far.
2. It would be helpful to include more recent studies, especially for the adenoviral and herpesviral vectors, to demonstrate ongoing advancements in these areas.
03. There are several minor grammatical issues and typos throughout the manuscript. A careful language edit will improve the overall readability of the manuscript. Ex: Line 109
Comments on the Quality of English Language
There are some minor grammatical issues and typos throughout the manuscript. A careful language edit will improve the overall readability and professionalism of the paper.
Round 2
Reviewer 1 Report
Comments and Suggestions for Authors
The manuscript has improved but is still not acceptable. There is a flood of reviews on ASFV and we do not need more review that cite each others unless they offer something really new. There are many little points that need correction but a major point remains that the immunogenicity of a viral protein in terms of inducing antibody or T cell responses is not an indicator that it could be a vaccine antigens. Expression of a viral protein in a viral vector and showing that you get an immune response is meaningless except that it shows that you vector works. Sometimes even mouse work is cited. We learn very little from such studies.The key question is if you see protection. By definition, all viral proteins if expressed in sufficient amounts will induce an immune response. This is because they are foreign to the host.
In addition, the level of immunology presented in the manuscript is too rudimentary. T helper cell are ignored; other functions of antibodies are ignored. The later is particularly important considering that neutralization is absent or poor following natural infection of ASFV. CTL's are mentioned for the viral proteins although very few studies really studied CTL.
Comments on the Quality of English Languagenot checked systematically
Author Response
Comments for the Author:
The manuscript has improved but is still not acceptable. There is a flood of reviews on
ASFV and we do not need more review that cite each other unless they offer something
really new.
Thank you for your valuable suggestions. We have changed the title of this review
into “Progress in African Swine Fever Live Vector Vaccine Development” to include
the bacterial and yeast vectors in addition to viral vectors. The whole manuscript has
been also changed accordingly.
Lines 13-15 in Abstract, “Live vector vaccines, a promising type of vaccine, offer
unique advantages and are a primary focus in ASFV vaccine research. This paper
focuses on the characteristics of live viral, bacterial and yeast vector vaccines”
The progress in development of ASFV bacterial and yeast vector vaccines was
added in Lines 562-610.
Lines 617-620 in Conclusions, “Bacterial and yeast vectors have also been
investigated to assess the immune responses elicited during the delivery of ASFV
antigens. However, challenge studies to evaluate the protective efficacy of some of
these vaccines in swine, the natural host of ASFV, have yet to be conducted.”
There are many little points that need correction but a major point remains that the
immunogenicity of a viral protein in terms of inducing antibody or T cell responses is
not an indicator that it could be a vaccine antigens. Expression of a viral protein in a
viral vector and showing that you get an immune response is meaningless except that it
shows that you vector works. Sometimes even mouse work is cited. We learn very little
from such studies. The key question is if you see protection. By definition, all viral
2
proteins if expressed in sufficient amounts will induce an immune response. This is
because they are foreign to the host.
Thank you for your valuable suggestions. We have thoroughly corrected this
manuscript. Particularly, we replaced the title of 3.1.1 with “Immunological properties
of antigens and corresponding immunological mechanisms”, and corrected all the
statements in the article that stated ‘The key factors for antigen selection are the
immunogenicity of the protein and the ability to induce specific T-cell immunity’.
Lines 121-144, we added the significance of antigenicity, immunogenicity, and the
ability to induce specific T-cell responses upon invasion by antigenic proteins.
Lines 185-216, we provided further insight into clarifying the functions of
antibodies, T-helper cells, and CTLs in immunity.
In addition, the level of immunology presented in the manuscript is too rudimentary. T
helper cell are ignored; other functions of antibodies are ignored. The later is
particularly important considering that neutralization is absent or poor following natural
infection of ASFV. CTL's are mentioned for the viral proteins although very few studies
really studied CTL.
Thank you for your valuable suggestions. We have added content in Lines 151-
216 related to the functions played by antibodies, T helper cells, and CTLs in immunity.
The neutralizing effects of antibodies, antibody-dependent cellular cytotoxicity
(ADCC), antibody-dependent cellular phag-ocytosis (ADCP), and complementdependent cytotoxicity (CDC), and the specific mechanisms of action of T helper cells
and CTLs were also described.
Reviewer 2 Report
Comments and Suggestions for Authors
The authors have not sufficiently addressed my comments raised during the first revision. They should have done it more creatively and in detail. Therefore, I would consider this work after extensive re-arrangement to the current study.
Author Response
Comments for the Author:
The authors have not sufficiently addressed my comments raised during the first
revision. They should have done it more creatively and in detail. Therefore, I would
consider this work after extensive re-arrangement to the current study.
Thank you for your valuable suggestions. We have revised the article according to
your suggestions.
(1) We added additional graphs (Figure 2) and charts (Table 2) to introduce the
genomic characteristics of vectors and the types of existing vectors that can provide
protection and their current status, respectively. We also revised Figure 1 to show
mechanism of host immune response induced by ASFV vector vaccine.
(2) We have gathered and assembled a comprehensive Table 2 of the vaccines
that have provided considerable protection in recent years and focused on their research
stage and whether they have practical applications;
(3) We provided a detailed elaboration on the benefits of adenovirus (lines 457-
461) and PRV virus (488-491). Furthermore, we incorporated bacterial and yeast
vectors to highlight their advantages compared with viral vectors (lines 562-610).
Therefore, we revised the title of the article and also introduced several types of
bacterial and yeast vectors, and analysed in detail the differences in their applications
compared with viral vectors.